# Remote loop evolution reveals a complex biological function for chitinase enzymes beyond the active site

Dan Kozome[1], Adnan Sljoka [2,3] & Paola Laurino [1,4] ✉

Loops are small secondary structural elements that play a crucial role in the emergence of new enzyme functions. However, the evolutionary molecular mechanisms how proteins acquire these loop elements and obtain new function is poorly understood. To address this question, we study glycoside hydrolase family 19 (GH19) chitinase—an essential enzyme family for pathogen degradation in plants. By revealing the evolutionary history and loops appearance of GH19 chitinase, we discover that one loop which is remote from the catalytic site, is necessary to acquire the new antifungal activity. We demonstrate that this remote loop directly accesses the fungal cell wall, and surprisingly, it needs to adopt a defined structure supported by long-range intramolecular interactions to perform its function. Our findings prove that nature applies this strategy at the molecular level to achieve a complex biological function while maintaining the original activity in the catalytic pocket, suggesting an alternative way to design new enzyme function.

Protein evolution is an essential process that drives diversity in nature by enabling the emergence of new functions[1]. Understanding how protein evolves is fundamental, not only for unraveling the natural diversification of proteins but also for designing new protein functions in vitro. The structure of more than 30,000 proteins reveals that nature uses a limited number of basic structures (scaffolds) to achieve an enormous variety of functions[2]. Indeed, many reports demonstrate that this diversity is often acquired by diversifying the flexible catalytic loop structures exposed on the protein's surface, while preserving the robust scaffold and the active site residues common among the enzyme family as core structures[3–5]. For instance, altering catalytic specificity by grafting loop[6] or shaping the catalytic activity via mutations in the catalytic loop[7]. These strategies are a major source of structural and functional variation within protein superfamilies[8]. Thus, mutations in the catalytic loop regions have garnered significant attention because they can directly impact function. Indeed, because of their functional roles like ligand binding[9], promoting function via conformational changes[10], and functional switching[11], the propensity of loop regions to acquire mutations is evolutionarily advantageous.

However, while previous studies have extensively explored the catalytic loops[6,7,12,13], the role of functional remote loop regions remains limited in functional switching or large conformational change allostery[14–17], especially, how new function is acquired via remote loop regions acquisitions/removals through evolution is poorly understood.

To understand how natural evolution acquires new functions via remote loops and its molecular mechanisms, we investigated the evolution of Glycoside Hydrolase family 19 chitinases (GH19 chitinase; EC 3.2.1.14). GH19 chitinase hydrolyzes the glycosidic bonds of chitin, namely beta-1, 4-linked N-acetyl-D-glucosamine. Chitin is the main component of exoskeletons and cell walls of various organisms, including arthropods and fungi. Despite lacking an endogenous substrate for plant chitinases, many plants synthesize various chitinases. One of the physiological roles of chitinase is to defend plants against pathogenic fungi by degrading chitin, a major component of the cell wall of many fungi[18]. Interestingly, even in GH19 chitinases, some chitinases do not exhibit antifungal activity. Taira et al. reported a difference in the antifungal activity of two chitinases from plants[19], - in this work, we will call them loopless and loopful GH19 chitinases.

[1]Protein Engineering and Evolution Unit, Okinawa Institute of Science and Technology Graduate University (OIST), Okinawa 904-0495, Japan. [2]Center for Advanced Intelligence Project, RIKEN, Tokyo 103–0027, Japan. [3]Department of Chemistry, York University, Toronto, ON M3J 1P3, Canada. [4]Institute for Protein Research, Osaka University, Suita, Japan. ✉e-mail: paola.laurino@oist.jp

Comparing the sequences and the structures of both GH19, the catalytic residues and the core structure that consists of the catalytic cleft are conserved. However, loopful GH19, a chitinase from rye seed, contains six loop regions, whereas loopless GH19, a chitinase from moss, lacks five of those loop regions seen in loopful GH19 (Fig. 1A, B). Interestingly, loopful GH19 exhibits antifungal activity against the fungi *Trichoderma sp.*, while loopless GH19 does not[19]. Mutational effects of insertion and deletions (InDels) of these loop regions on catalytic activity or fold stability are previously reported[20–24]. However, the molecular evolutionary basis for chitinase achieving an increase in antifungal activity while seemingly having no impact on catalytic activity remains unclear.

Herein we investigate the evolution of GH19 chitinase to understand how functional innovation occurred. We infer five ancestral proteins with different loop combinations in remote loop regions. By characterizing these proteins variants, we discover that loop II is critical for acquiring antifungal activity. Using structural analysis, molecular dynamics (MD) simulations, and computational studies, we show that this remote loop allows new functions through fine-tuned intramolecular interactions. Imaging studies via fluorescence microscope experiments unravel that

the acquisition of remote loop II access the substrate in a new specific cellular location, the fungal cell wall. Our study highlights the molecular evolutionary basis of how nature acquired remote loops in a protein structure to access a complex biological function.

## Results

### Phylogenetic analysis reveals the emergence of antifungal activity in GH19 chitinase

To identify the remote loop regions acquisition that played a role in the emergence of antifungal activity and when such an acquisition occurred in GH19 chitinase, we performed phylogenetic analysis and ancestral sequence reconstruction. After extensive collection of all available sequences belonging to GH19 chitinase from plants and the removal of redundancy, we inferred the maximum-likelihood phylogenetic tree of 179 GH19 chitinase sequences. We used ancestral protein reconstruction to infer the most likely amino acid sequence for each ancestral node in the phylogeny. To avoid the bias that phylogenetic classification is affected by the presence or absence of loop regions, we inferred two phylogenetic trees using a multiple sequence alignment (MSA) of the collected sequences, and a MSA modified by

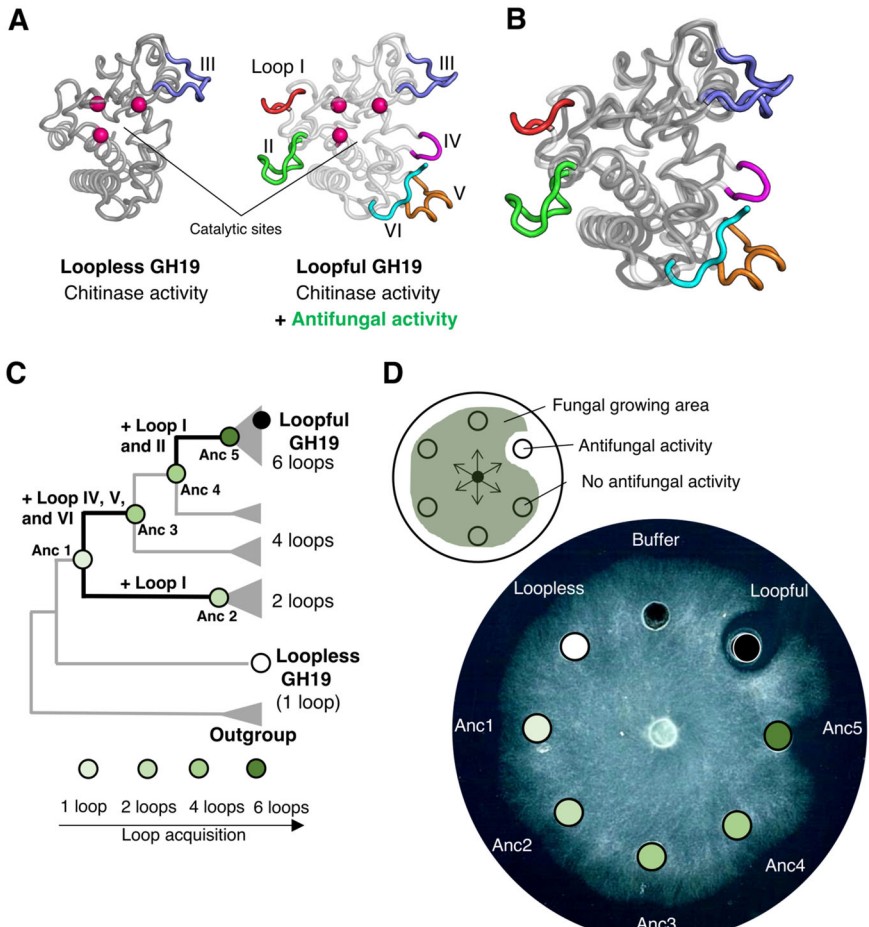

**Fig. 1 | Antifungal activity is acquired during evolution in GH19 chitinase.**
**A** Structural comparison of two types of GH19 chitinase, loopless and loopful. They share the core region (darker and lighter gray in loopless and loopful GH19, respectively), two catalytic glutamic acids, and one serine that holds catalytic water (magenta spheres) but differs in loop regions, named I to VI (shown in red, green, light blue, purple, orange, and cyan, respectively). **B** Structural superimposition of loopless and loopful. The detailed catalytic mechanism that they conserve is shown in Supplementary Fig. 1. **C** Schematic representation of a phylogenetic tree of GH19 chitinase. The five ancestral nodes that were characterized are labeled and colored according to the number of loop regions (80% lighter green, one loop; 60% lighter

green, two loops; 40% lighter green, four loops; 25% darker green, six loops). Evolutionary transitions containing loop acquisitions are highlighted with thick black lines. A multiple sequence alignment of two modern and five ancestor sequences and a full phylogenetic tree are shown in Supplementary Figs. 2 and 3, respectively. **D** Top, a schematic representation of fungal hyphae expansion inhibition assay against Trichoderma longibrachiatum as the test fungus. Bottom, the assay plate. Each well contains 10 µL of sterilized buffer (10 mM sodium acetate buffer, pH 5.0) or 500 pmol of protein samples dissolved in the same buffer. Experiments were performed three times independently with similar results.

## Table 1 | Enzymatic activities and melting temperature of the two modern and five ancestral GH19 chitinases

| Protein | Loop | Specific activity (U$^a$/ mol) × 10$^9$ | $T_m$ (°C) |
|---|---|---|---|
| Anc1 | III | 1.44 | 58.5 |
| Anc2 | I and III | 0.75 | 52.0 |
| Anc3 | III, IV, V, and VI | 1.20 | 72.0 |
| Anc4 | III, IV, V, and VI | 1.05 | 72.5 |
| Anc5 | I, II, III, IV, V, and VI | 0.85 | 66.0 |
| Loopless | III | 1.51 | 68.0 |
| Loopful | I, II, III, IV, V, and VI | 0.83 | 62.5 |

$^a$One unit of activity is defined as the enzyme activity that produced one μmol of GlcNAc per minute at 37 °C. Melting temperature ($T_m$) was measured using differential scanning fluorimetry (DSF) followed by the procedures as described in the Methods.

removing the six loop regions in question. After we confirmed that the tree topology did not change based on the presence/absence of loop regions in the MSA (Supplementary Figs. 2, 3, and 4), we used the tree based on the unmodified MSA and selected five ancestral nodes where InDels of the loop regions occurred, designated Anc1 to Anc5 based on the presence/absence of loop regions in the extant sequences (Fig. 1C). We experimentally characterized chitinase activities and antifungal activities of five reconstructed ancestral proteins and two extant GH19 sequences. All five ancestral proteins showed a similar level of chitinase activity to the two modern GH19 chitinases (Table 1), while Anc5 also exhibited antifungal activity. Anc1-4 did not exhibit antifungal activity (Fig. 1D), suggesting that antifungal activity seems to be acquired during the transition from Anc4 to Anc5.

### Characterization of the functional intermediates between Anc4 and Anc5

To identify the key loop insertion for antifungal activity acquisition, we constructed six variants of two reconstructed ancestral proteins, Anc4 and Anc5 with different combinations of loop I and II (Anc4 + Loop I, Anc4 + Loop II, Anc4 + Loop I and II, Anc5ΔLoop I, Anc5ΔLoop II, and Anc5ΔLoop I and II). We experimentally characterized antifungal activities and hydrolytic activities of these variants. To compare the strength of their antifungal activities, we calculated IC$_{50}$ by performing a hyphal re-extension inhibition assay. Addition/removal of loop I and/or II regions did not affect their hydrolytic activities (1.1–1.5-fold) except for in the case of loop I addition to Anc4. Insertion of loop I to Anc4 resulted in a loss of catalytic activity of Anc4 (Fig. 2B). However, removing loop II from Anc5 decreased its antifungal activity by 12-fold while removing loop I did not influence its antifungal activity (roughly 1.02-fold increase, Fig. 2B). These results strongly suggest that loop II has a role in enhancing antifungal activity. However, it is worth noting that the addition of loop II to Anc4 did not improve its antifungal activity. Anc4 and 5 differ by 45 substitutions (Fig. 2A, C), suggesting that loop II enhances antifungal activity in combination with residues in the protein scaffold.

### Structural analysis of Anc4 and Anc5 and molecular dynamics (MD) simulations revealed the importance of long-range interactions in the emergence of antifungal activity

To get structural insights into the emergence of antifungal activity in GH19 chitinase, we first solved the X-ray crystal structure of Anc4 and Anc5 (Supplementary Table 4). Overall, the backbone structures of Anc4 and Anc5 were nearly identical (RMSD of Cα = 0.478 Å) and the orientation of catalytic residues is similar, suggesting that loop I and II addition/removal did not cause major structural disruptions. This is supported by the fact that addition/removal of loop I and/or II regions did not affect their hydrolytic activities except for the loop I insertion into Anc4 (Fig. 2B). The loss of the hydrolytic activity in Anc4 via loop I

insertion might be due to the lack of disulfide bonding between loop I and the core scaffold. This is because the cysteine residue that forms the disulfide bonding with loop I is replaced by another residue (Val 81 in the MSA, Supplementary Fig. 5).

Although Anc4 and Anc5 showed no significant structural difference and their mutants Anc4 + Loop II and Anc5ΔLoop I had similar hydrolytic activity, their antifungal activities were significantly different (Anc5ΔLoop I showed 56-fold higher activity than Anc4 + Loop II, Fig. 2B). Therefore, we explored the contributions of protein dynamics to the acquisition of antifungal activity by performing MD simulations. We performed four runs of 200 ns simulations of the model structure of Anc4 + Loop II and Anc5ΔLoop I built using Anc4 and Anc5 structures (PDB 8HNE and 8HNF). In Anc5ΔLoop I, the simulation showed reduced mobility of the loop II regions and a loop region between the ninth and tenth α-helices (positions 192–201) compared to Anc4 + Loop II (Fig. 3A, B). Some of the 45 substitutions support the stabilization of loop II in Anc5ΔLoop I. Structural comparison of Anc4 and Anc5 revealed six substitutions forming new interactions with loop II regions (Fig. 3C). Two substitutions, p12K and n13H (small and large character indicates Anc4 and Anc5 state, respectively), formed new hydrogen bonding with Asp73 residue in loop II. The His 13 residue is also involved in hydrophobic interaction with Trp78 residue in loop II (Fig. 3C, upper left in the enlargement). d197(217)R (residue numbers are responsible to Anc5ΔLoop I; residue numbers in Anc5 state are in parentheses) substitution formed new hydrogen bonds between the N terminus and the tenth α-helix (Fig. 3C, upper right in the enlargement). n193(213)G substitution reduced structural hindrance with the Tyr76 residue in loop II (Fig. 3C, bottom left in the enlargement). Loop II insertion into Anc4 caused a change in conformational orientation of Ser 58, leading to a formation of a hydrogen bond between the oxygen atom of Ser 58 and the nitrogen atom of the Gly 60 as seen between Thr 65 and Gly 67 in Anc5. y194(214)F substitution reduced structural hindrance with the oxygen atom of Thr 65 (Fig. 3C, bottom right in the enlargement). These substitutions are important for stabilizing the loop II region to perform antifungal activity. Furthermore, to investigate if the flexibility of the loop region between the ninth and tenth α-helices (positions 192–201) affects the antifungal activity, we also constructed the mutant of Anc5 and Anc5ΔLoop I that only disrupts the stability of this loop. MD simulations and antifungal activity of these mutants revealed that the flexibility of a loop region between the ninth and tenth α-helices (positions 192–201) did not contribute to enhance antifungal activity (Supplementary Fig. 5 and Supplementary Table 6).

To assess whether residues in the scaffold have long-range effects on rigidity and conformational modifications of loop II, we utilized rigidity-transmission allostery (RTA) algorithms[25]. RTA is a computational approach based on mathematical rigidity theory[26,27] and graph theory, which analyzes long-range communication and allosteric networks within protein structures[28–31]. RTA measures whether local mechanical perturbation of rigidity at one site propagates and modifies rigidity and conformational degrees of freedom at distant site(s) in protein structure. Starting with a structure, RTA first utilizes the method Floppy Inclusion and Rigid Substructure Topography (FIRST)[32] to generate a constraint network, where protein structure is modeled in terms of vertices (atoms) and edges (i.e., covalent bonds, electrostatic bonds, hydrogen bonds, and hydrophobic contacts). Every potential hydrogen bond is ranked and assigned energy strength according to its donor-hydrogen-acceptor geometry. Upon rigidification of individual site(s) (i.e., residues), RTA then quantifies transmission (changes) of degrees of freedom and strength of communication across protein structure. RTA analysis on Anc5ΔLoop I showed that several residues in the scaffold are involved in long-range communication with loop II, revealing an allosteric network of residues that transmit communication with loop II (Fig. 3D). Interestingly, many of the substitution residues are part of this communication network, suggesting they impact the stability and conformational dynamics of loop II. To further probe

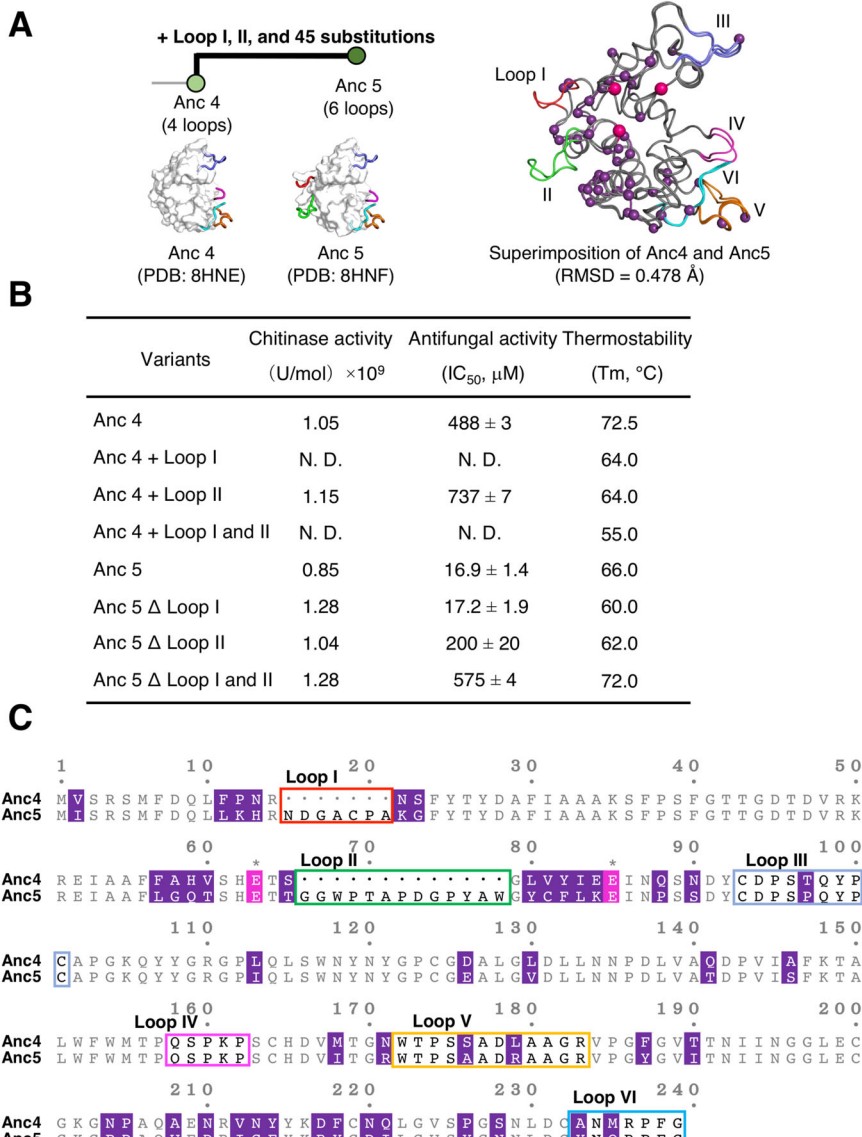

**Fig. 2 | Characterization of eight variants of two ancestral proteins reveal the key remote loop for the acquisition of antifungal activity. A** Left, a schematic representation of an evolutionary transition from Anc4 to Anc5. Crystal structures of Anc4 and Anc5 solved to 1.13 Å and 1.57 Å, respectively. Right, structural superimposition of Anc4 and Anc5. Two catalytic glutamic acids and one serine reside that holds catalytic water are indicated as magenta spheres. 45 substitution residues are indicated as purple spheres. **B** Summary of the effects of loop presence/absence on the hydrolytic and antifungal activities. One unit of activity is defined as

the enzyme activity that produced 1 μmol of GlcNAc per minute at 37 °C. ±indicates standard deviation between three replicates. N.D. indicates activity not detected. Melting temperature ($T_m$) was measured using differential scanning fluorimetry followed by the procedures as described in the Methods. **C** An alignment of Anc4 + Loop II and Anc5ΔLoop I sequence. Loop regions II to VI are highlighted in green, light blue, purple, orange, and cyan squares, respectively. Two glutamate residues and one serine residue are shown in magenta. 45 substitutions residues are shown in purple.

this, we applied FIRST and decomposed the protein structures into rigid and flexible regions. FIRST rigid cluster decomposition on Anc4 + Loop II and Anc5ΔLoop I structures predicted that loop II is stabilized by intra-hydrogen bonding between the Pro 72 and the Asp 73 (residue number is based on the MSA in Fig. 2), which is observed only in Anc5ΔLoop I structure (Supplementary Fig. 6). This results in a strong rigid cluster in Anc5ΔLoop I which persists over wide hydrogen bond energy strengths (Supplementary Fig. 6).

## Loop II leads to gain of antifungal activity by promoting binding to the fungal cell wall

To understand the complex biological role of loop II, we performed fluorescence microscope experiments. These experiments aimed at verifying if the acquisition of loop II in GH19 chitinase plays a role in binding to the surface of the fungal cell wall. For this purpose, we

prepared the catalytically inactive mutants of Anc4, Anc4 + Loop II, Anc5, Anc5ΔLoop I, Anc5ΔLoop II, and Anc5ΔLoop I and II by replacing catalytic glutamate residue with glutamine (The Glu 67 in the MSA, Supplementary Fig. 2). This mutation makes proteins lose chitin degrading activity in the fungal cell wall. These inactive mutants were tagged with AlexaFluor488. By fluorescence microscope experiments, we observed that only Anc5 and Anc5ΔLoop I bound to the surface of fungal hyphae (Fig. 4A, B and Supplementary Fig. 7). This result is consistent with the result that Anc5 and Anc5ΔLoop I showed 28-fold stronger antifungal activity than Anc4 (Fig. 2B), suggesting that the acquisition of loop II is necessary to access the substrate in the cell wall of fungi to perform antifungal activity. Surprisingly, Anc4 + Loop II did not exhibit any binding activity to the fungal cell wall, despite the presence of Trp residues (positions 68 and 78 in the MSA, Fig. 2B) known for their involvement in substrate binding interactions[33], as well as in

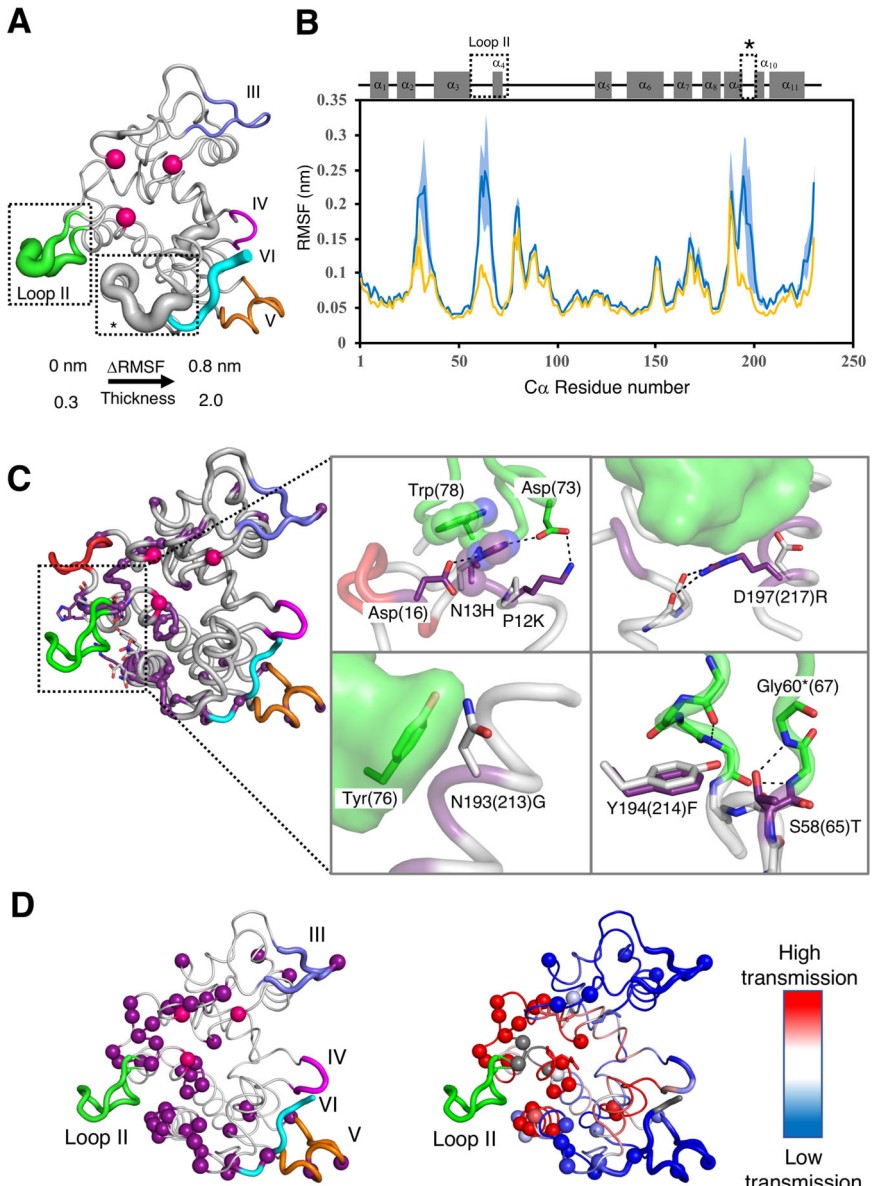

**Fig. 3 | Comparison of Anc4 + Loop II and Anc5ΔLoop I exhibit the structural and dynamics contribution of Loop II and scaffold substitutions. A** The difference in Δroot mean square fluctuation (ΔRSMF = RSMF of Anc4 + Loop II − RMSF of Anc5ΔLoop I) is mapped on the structure of Anc5ΔLoop I as the thickness of the cartoon representation. Spheres indicates two catalytic glutamic acids and one serine that holds catalytic water. **B** Plots of the RMSF of Cα of each residue in Anc4 + Loop II (blue) and Anc5ΔLoop I (orange). Error bars are indicated as shades (*n* = 4). A schematic representation of secondary structures of GH19 chitinase is shown. Dashed squares indicate the region where ΔRSMF is more than 0.05 nm. RMSDs of the Cα atoms of all residues are shown in Supplementary Fig. 5B. **C** Cartoon_tube representation of Anc5. Sphere representation indicates 45 substitution residues (purple) and two catalytic glutamic acids and one serine that holds catalytic water (magenta). Resides around 4 Å from loop II are shown in sticks. Intramolecular interactions stabilizing loop II regions are shown in the enlargements. Resides in Anc4 and Anc5 state are shown in gray and purple sticks, respectively. Black dashed lines indicate hydrogen bonds. Residue numbers are based on Anc4. In the case that the number shifts due to loop insertions, residue numbers of Anc5 state are in parentheses. An asterisk indicates the number of Anc4 + Loop II. Loop regions I to VI are shown in red, green, light blue, purple, orange, and cyan, respectively. **D** Left, computational analysis of long-range communication in Anc5ΔLoop I. Cartoon representation of Anc5ΔLoop I with cartoon_tube representation of loop II to VI (shown in green, slate, purple, orange, and cyan, respectively). Sphere representation indicates 45 substitutions residues (purple) and two glutamic acids and one serine that holds catalytic water (magenta). Right, rigidity-transmission allostery (RTA) communication analysis on Anc5ΔLoop I. Residues are colored based on the intensity (red being highest) of long-range rigidity-transmission communication with loop II (green). Spheres indicate 45 substitutions.

stabilizing Loop II (Figs. 2B and 4A)[20]. This result suggests that Loop II is necessary to have a potential binding activity to fungal cell wall but requires the additional substitutions to perform its binding activity.

## A gain of new function through a remote loop acquisition in combination with substitutions

To investigate the contribution to the antifungal activity of the 45 substitutions, we constructed mutants of Anc4 + Loop II with

substitutions in proximity to loop II. We chose six substitutions within 4.0 Å from Loop II and introduced them accordingly to their proximity to each other's (Fig. 4C, D: mutations A, p12K and n13H; mutation B, s58T mutations C, n193G, y194F, and d197R). Mutations A, B, and C increased antifungal activity 4.7, 1.2, and 5.4-fold, respectively. These substitutions increased antifungal activity while retaining the same degree of chitinase activity (Table 2). Cell wall binding activity assay confirmed that integrating these mutations increased binding activity

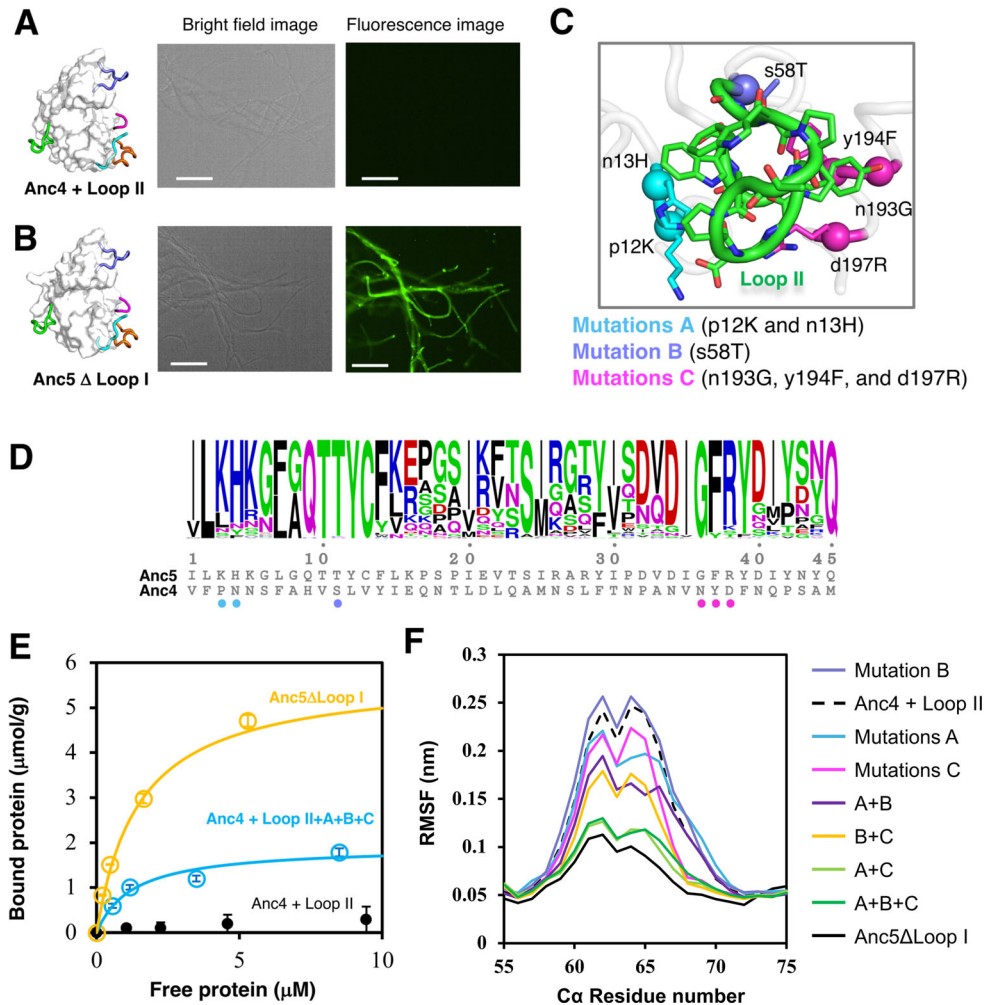

**Fig. 4 | Gain of antifungal activity through substitutions that rigidify loop II to bind fungal cell wall. A** From left to right: the structure of Anc4 + Loop II; Bright field image of fungal hyphae; Fluorescence microscope image shows no binding to the surface of fungal hyphae. **B** From left to right: the structure of Anc5ΔLoop I; Bright field image of fungal hyphae; Fluorescence microscope image shows Alexa Fluor 488-labeled Anc5ΔLoop I binding to the surface of fungal hyphae. Each 50 μL of 2 μM Alexa Fluor 488-labeled proteins Anc4 + Loop II and Anc5ΔLoop I is mixed with *T. longibrachiatum* hyphae in 20 mM sodium phosphate buffer, pH 7.4 at 25 °C. Images were captured after washing excess of the labeled proteins. All scale bares are 100 μm. Experiments were performed three times independently with similar results. **C** The crystal structure of Anc4 + Loop II integrated with all six substitutions (p12K, n13H, s58T, n193G, y194F, and d197R) revealed the interactions of these residues with loop II. Substitutions were introduced gradually due to the convenience experimentally as mutations A (cyan), p12K and n13H; mutation B

(slate), s58T mutations C (magenta), n193G, y194F, and d197R. Small and large characters indicate the Anc4 and Anc5 states, respectively. Side chains and Cα atoms of the introduced residues are shown as stick and spheres, respectively. **D** Sequence logo of 45 positions where substitutions occurred between Anc4 and Anc5. The height of logo represents the frequency of amino acid residues at each position among loopful-type GH19 chitinases. **E** Binding curves of Anc4 + Loop II + A + B + C and Anc5ΔLoop I. Binding curves were fitted with plots of the absorbance of the supernatants of each protein samples at 280 nm after pull-down assay. The pull-down assay was performed in 10 mM sodium acetate buffer (pH 5.0) at 25 °C, using 0.1% (w/v) cell wall fraction from T. *longibrachiatum* as the substrate. Error bars represent the SD of the means (*n* = 3). **F** RMSFs of the Cα atoms of Loop II residues in Anc4 + Loop II, Anc5ΔLoop I, and mutants of Anc4 + Loop II. RMSDs of the Cα atoms of all residues are shown in Supplementary Fig. 5B.

to the fungal cell wall (Fig. 4E). MD simulations of these mutants revealed a gradual reduction of loop II flexibility directly related to the introduction of these six substitutions (Fig. 4F) Furthermore, these substitutions decreased only the mobility of loop II not the one of a loop region between the ninth and tenth α-helices (positions 192–201, Supplementary Fig. 5). Furthermore, crystal structures of Anc4 + Loop II + A and Anc4 + Loop II + A + B + C revealed that their loop II structures are retained as in the structure of Anc5's loop II, namely the antifungal active variant (Supplementary Figs. 8 and 9). Altogether, these results exemplify that antifungal activity enhanced by cell wall binding activity is supported by the substitutions that rigidify loop II. This demonstrates the molecular evolution trajectory of acquiring a new functional loop, wherein a distant loop is incorporated through Insertions and Deletions (InDels). This process is further

complemented by accumulated substitutions, leading to the protein gaining a novel function.

## Discussion

Proteins use the limited number of folds and have evolved their structure and function using substitutions and InDels of loops and other secondary structures[34,35]. However, most studies focused on substitutions, and InDels are rarely considered. In these rare cases, when InDels are studied, they consist of one to three amino acid removal or addition[36,37] or graft of the catalytic loops[6,7,12]. Shining light on the roles of insertion or deletions of entire remote loop regions in protein evolution can provide further understanding of how we can engineer new protein functions. Our study reveals the molecular mechanism of an enzyme, through addition of a remote loop, can

**Table 2 | Summary of enzymatic and binding parameters of Anc4 + Loop II, its mutants, and Anc5ΔLoop I**

| Variants | Mutations | Hydrolytic activity (Uᵃ/mol) × 10⁹ | Antifungal activity (IC₅₀) | Binding activity Bmax (μmol/g) | Kd (μM) |
|---|---|---|---|---|---|
| Anc4 + II | N/A | 1.15 | 737 ± 7 | 0.57 ± 0.20 | 8.93 ± 5.46 |
| Anc4 + II + A | n13H/p12K | 1.09 | 155.9 ± 13 | n.d | n.d |
| Anc4 + II + B | s58T | 1.14 | 593 ± 22 | n.d | n.d |
| Anc4 + II + C | d197R/y194F/d197R | 1.32 | 136.2 ± 4.3 | 1.66 ± 0.61 | 4.31 ± 2.06 |
| Anc4 + II + A + B | | 0.79 | 129.3 ± 5.7 | n.d | n.d |
| Anc4 + II + A + C | | 1.44 | 128.8 ± 7.4 | 3.61 ± 0.29 | 3.33 ± 0.71 |
| Anc4 + II + B + C | | 1.43 | 72.6 ± 10 | 0.76 ± 0.29 | 1.75 ± 0.33 |
| Anc4 + II + A + B + C | | 1.43 | 63.43 ± 3.8 | 2.04 ± 0.04 | 1.59 ± 0.07 |
| Anc5ΔI | 45 subs | 1.28 | 17.2 ± 1.9 | 5.93 ± 0.19 | 1.47 ± 0.11 |

ᵃOne unit of activity is defined as the enzyme activity that produced one μmol of GlcNAc per minute at 37 °C. ± indicates standard deviation between three replicates. n.d. indicates activity not detected.

acquire a new distinct function distant from the catalytic site while maintaining its original activity.

To address this, we first identified the transition where the remote loop regions were inserted or deleted during evolution of the enzyme. This demonstrates that phylogenetic analysis and ancestral sequence reconstruction can explore remote loop acquisition in enzymes and thus identify potential hotspots for loop engineering. Most studies using ancestral sequence reconstruction have been performed with a fixed length of MSA due to the ambiguous evolutionary information of InDels. In our case, GH19 chitinases acquire an additional antifungal activity over original chitinase activity, through InDels of remote loop regions. Our phylogenetic analysis with two MSAs (with/without the considered loop regions) showed that InDels of remote loop regions did not occur frequently and the tree topology did not change significantly due to the InDels of remote loop regions (Supplementary Fig. 3). Thus, we identified the ancestral nodes where the protein acquired/lost remote loops and inferred functional ancestral proteins with different loop combinations (InDels). As previous studies[37–39] demonstrated that InDels in loop regions are highly tolerated, our ancestral proteins showed robustness to add/remove loops. Thus, we demonstrate that ancestral sequence reconstruction approach in enzyme family with structural variations in loop regions is useful for designing enzyme with different loop combinations and exploring the potential remote site for loop engineering.

Surprisingly these ancestral proteins have the same sequence length and almost identical structure (RMSD of Cα = 0.478 Å), however, they showed significantly different antifungal activity (Anc4 + Loop II and Anc5 ΔLoop I differ 40-fold in antifungal activity in addition to 45 substitutions in the scaffold), indicating that structural dynamics play an important role for the additional functions[40]. Our mutational analysis revealed that some of substitutions are conserved in loopful-type GH19 chitinases (Fig. 4D) and six substitutions are needed to perform new function (Table 2). In the evolution of GH19 chitinase, loop insertion comes with stabilizing the key remote loop regions is important to perform new protein functions, and this stabilization requires additional substitutions (Fig. 4 and Table 2). By comparing the evolutionary consequences of homologous proteins, researchers have discovered that loop regions, or regions decorated by loops, govern protein dynamics to execute distinct functions[3,41]. This characteristic can be harnessed through protein engineering to achieve functional switches[2,6,14]. These significant mutational steps occur alongside substitutions in natural sequences[42]. To date, no experimental work has demonstrated that such a dramatic functional shift can be achieved remotely from the catalytic site. We unveiled the missing piece: a direct evolutionary pathway for functional emergence involving the insertion of a crucial remote loop, followed by subsequent substitutions that stabilize the dynamics of this critical loop.

In the evolution of GH19, it is interesting to note that the enzyme became dual-functional with a new complex biological activity, accessing to the fungal cell wall while maintaining its original activity even though a trade-off between the catalytic activities to original substrate and the promiscuous substrate is often observed in protein evolution[43,44]. Trade-off can be associated with a change in the conformational dynamics of the catalytic pocket optimizing to the substrate[40], and bifunctionality can be achieved when this trade-off is weak enough to maintain original activity[45]. In the case of GH19, there is no significant change in the dynamics near the catalytic residues due to the mutations and original catalytic activities retained (Figs. 2B, 3A, and Supplementary Fig. 5). The enzyme appears to possess an inherent conformer capable of displaying antifungal activity, but this conformation is only accessible when the protein interacts with the fungal cell wall, its substrate. The inserted remote loop played a pivotal role in enabling remote functionality, thanks to complementary substitutions within the protein scaffold. Thus, protein can acquire additional function while retaining its original catalytic activity.

Strikingly, our findings show that the key loop insertion allowed the accessibility to the substrate in a different cell types and cellular location (Fig. 4A, B). Since the target substrate for antifungal activity is located in the cell wall of fungi, not in the solution, the protein needs to access the substrate location to perform protein function. The fluorescence labeling experiments clearly show that chitinase access to the fungal cell wall to acquire antifungal activity (Figs. 2B, 4, and Supplementary Fig. 7). Nevertheless, the acquisition of this new and complex function is possible only through the long-range interaction between the inserted secondary element and the substituted residues in the protein scaffold (Figs. 3, 4, and Supplementary Figs. 5 and 8). Herein we demonstrate the molecular basis of how remote loop insertion plays an important role in accessing the substrate.

In addition, our findings are key to developing protein engineering methods aimed at accessing water-insoluble substrates such as cellulose, chitin, and plastic. To achieve efficient degradation of water-insoluble substrates, accessibility is equally important to catalytic efficiency[46]. Furthermore, we demonstrate how functional peptide grafting is not only limited to catalytic loop, but it can be extended to remote loops. In conclusion, we show an alternative and effective way to redesign or expand enzyme function, opening new ways of thinking for enzyme designers.

## Methods

### Phylogenetic analysis and ancestral sequence reconstruction

2617 sequences of plant GH19 chitinase were collected from the UniProtKB database[47]. Redundant sequences with more than 50% sequence identity were filtered using the CD-HIT program. The resulting 682 sequences were aligned using MAFFT ver. 7[48]. Only GH19 catalytic domain sequences were aligned, and additional domains were

manually removed using an alignment visualizing software, Aliview[49]. Sequences of a loopless type GH19 chitinase from *Gemmabryum coronatum* (Uniprot: A9ZSX9; residues 25-228) and a loopful type GH19 chitinase from *Secale cereale* (Uniprot: Q9FRV0; residues 24-266) were used as references for the smallest and largest GH19 catalytic domain, respectively. 11 bacterial GH19 chitinase sequences were added to the dataset as an outgroup. The resulting 179 sequences were aligned with MAFFT and a maximum-likelihood phylogenetic tree was estimated using the model (WAG + F + I + G4) automatically determined in IQ-TREE[50]. Ancestral protein sequences were reconstructed using the empirical Bayesian method applied by IQ-TREE[50]. The ancestral sequences Anc1 to Anc5 were reconstructed using the WAG substitution matrix together with the maximum-likelihood phylogenetic tree.

## Cloning and site-directed mutagenesis

Codon-optimized genes encoding the ancestral GH19 chitinase proteins and loopful type GH19 chitinase from *Secale cereale* (Uniprot: Q9FRV0; residues 24-266) were synthesized by TWIST Bioscience and cloned into the pET-22b (+) vector using the iVEC3[51]. PCR amplifications for synthetic genes and a linear-pET-22b (+) vector were performed with using PrimeSTAR® Max DNA polymerase (TaKaRa) and the designed primers (Supplementary Table 1) containing appropriate overlapping regions for iVEC3. The gene coding loopless type GH19 chitinase from *Gemmabryum coronatum* (Uniprot: A9ZSX9; residues 25-228) cloned into the pET-22b (+) vector was a gift from Toki Taira.

Site-directed mutagenesis was achieved by Inverse PCR using PrimeSTAR® Max DNA polymerase (TaKaRa) with the designed primers (Supplementary Table 1). Successful cloning and mutagenesis were confirmed by Sanger sequencing.

## Protein expression and purification

*E. coli* Shuffle T7 (DE3) cells harboring the protein gene of interest were grown in LB at 37 °C to $OD_{600}$ 0.6–0.8, induced with 0.1 mM β-d-1-isopropyl thiogalactopyranoside and incubated further for 24 h at 18 °C. Cells were pelleted and stored at −80 °C before protein purification. The cells were disrupted by sonication in a 20 mM Tris-HCl buffer, pH 8.0. The sonicated extract was separated into soluble and insoluble fractions by centrifugation at $12,000 \times g$ for 15 min at 4 °C. The soluble fraction was dialyzed against 10 mM sodium acetate buffer, pH 5.0, and filtered before applying to a RESOURCE Q column (6 mL, Cytiva) or HiTrap SP HP column (5 mL, Cytiva) previously equilibrated with the same buffer. The elution was done with a linear gradient of NaCl from 0 to 0.3 M in the same buffer. The recombinant protein fractions were collected and dialyzed against a 5 mM Tris-HCl buffer containing 150 mM NaCl, pH 8.0. Purified recombinant proteins were concentrated using a Millipore centrifugal protein concentration device (10 kDa cutoff) and loaded onto a Superdex200 Hiload 16/600 column (Cytiva) equilibrated with 5 mM Tris-HCl buffer containing 150 mM NaCl, pH 8.0. Protein purity was confirmed by SDS−PAGE, and protein concentrations were measured spectrophotometrically using molar absorption coefficients calculated in ProtParam (http://expasy.org/tools/protparam.html).

## Chitinase activity assay

Chitinase activity was measured colorimetrically with glycol chitin as a substrate. 10 μL of the sample solution was added to 150 μL of 0.2% (w/v) glycol chitin solution in 0.1 M sodium acetate buffer, pH 5.0. After incubation of the reaction mixture at 37 °C for 15 min, the reducing power of the mixture was measured with ferric ferrocyanide reagent by Imoto and Yagishita[52]. One unit of activity was defined as the enzymatic activity that produced 1 mmol of GlcNAc per minute at 37 °C.

## Antifungal activity assay

**Qualitative assay.** An antifungal assay was performed according to the method of ref. 18 with modification. An agar disk (4 mm in diameter) containing the test fungus, *T. longibrachiatum*, prepared from the cultured fungus on potato dextrose broth with 1.5% (w/v) agar (PDA), was placed in the center of a Petri dish containing PDA. Wells were subsequently punched into the agar at a 15 mm distance from the center of the Petri dish. 500 pmol of each protein sample was placed into the wells. The plates were incubated for 24 h at 25 °C.

**Quantitative assay.** Hyphal re-extension inhibition assay was done by using *T. longibrachiatum*. Agar disks (4 mm and 1 mm in diameter and in-depth, respectively) containing the fungal hyphae, which were derived from the resting part of the fungus previously cultured on potato dextrose broth containing 1.5% (w/v) agar (PDA), were put on another PDA plate with the hyphae attached side down. 5 μL of sterile water or sample solution were overlaid on the agar disks, and then the plate was incubated at 25 °C for 12 h. After incubation, images of the plates were scanned using an image scanner. The areas of the re-extended hyphae were calculated as numbers of pixels by GNU Image Manipulation Program (GIMP, ver. 2.0). The protein concentration required for inhibiting the growth of the fungus by 50% was defined as $IC_{50}$ and determined by constructing dose-response curves (percentage of growth inhibition versus protein concentration).

## Cell wall binding assay

The binding activity of protein to the fungal cell wall was measured by pull-down assay followed by a previous method with modifications (ref.). 5 μL of $10^6$ spores/mL of T.*longibrachiatum* was added to 1 L of sterilized PDB (Potato dextrose broth) in a 3 L flask and was incubated at 30 °C for 48 h with shaking at 300 rpm to obtain the mycelia of the fungi. The mycelia were collected by filtration, washed extensively with water, and lyophilized. 1.5 g of the dried mycelia were incubated three times in 50 mL of 1 M NaOH at 65 °C for 30 min. The alkali-insoluble pellet was washed five times with water and twice with 10 mM of sodium acetate buffer (pH 5.0) and lyophilized and stored as cell wall fraction.

250 μL of 0.2% (w/v) cell wall fraction suspension prepared in 10 mM sodium acetate buffer (pH 5.0) was mixed with an equal volume of the protein solution. The mixture was incubated at 25 and shaked with 1000 rpm for 1 h. After shaking, the reaction mixture was centrifuged at $12,000 \times g$ for 20 min. The protein concentrations of the supernatant were measured with the absorbance at 280 nm. The $K_d$ and $B_{max}$ (μmol/g of cell wall fraction) were determined by fitting the binding isotherms to a one-site binding equation, where $P$ represents protein; $[P_{bound}] = B_{max} [P_{free}]/(K_d + [P_{free}])$. The fitting was done via nonlinear regression analysis.

## Differential scanning fluorimetry

Differential scanning fluorimetry experiments were performed using StepOnePlus Instrument, a real-time PCR equipment. Reaction mixtures contained 2.5 μM protein in 10 mM sodium acetate buffer, pH 5.0, 5× SYPRO orange dye in a total volume of 20 μL and dispensed into a 96-well PCR plate. Fluorescence intensities were monitored continuously as the sample mixtures were heated from 20 °C to 99 °C at a rate of 1% (approximately 1.33 °C/min), using the ROX channel. Melting temperatures were determined by fitting the fluorescence derivative data to a quadratic equation in the vicinity of the Tm in R software.

## Crystallization

After purification, Anc4 and Anc5 were concentrated at 10.0 mg/ml and 5.67 mg/ml in the 5 mM Tris-HCl buffer, pH 8.0, respectively. Anc4 + Loop II + A and Anc4 + Loop II + A + B + C were concentrated at 6.00 mg/ml and 3.80 mg/ml in the 10 mM sodium acetate buffer containing 150 mM NaCl, pH 5.0, respectively. Initial crystallization screens were performed using various crystallization screening kits commercially available. The protein solution drop (400 nL) was mixed with 400 nL of a reservoir solution and then equilibrated with 50 μL of

the reservoir solution using a crystallization robot, Mosquito® Xtal3. The crystallization conditions were screened using the sparse-matrix sampling method, according to the sitting-drop vapor diffusion method at 20 °C in a 96-well plate (violamo). After a week, well-formed crystals of Anc4 and Anc5 were obtained using PEGRx 1 (Hampton Research) and further optimized to the condition (15% and 27% (w/v) polyethylene glycol monomethyl ether 2000, 0.1 M MES monohydrate pH 6.0 for Anc4 and Anc5, respectively). Well-formed crystals of Anc4 + Loop II + A and Anc4 + Loop II + A + B + C were obtained using Eco-screen (Molecular Dimensions) at the condition (25% (w/v) polyethylene glycol 3350, 0.1 M Bis-Tris, pH 6.5).

## X-ray data collection

For data collection, the crystals of Anc4 and Anc5 were soaked in cryoprotectant buffer (40% (w/v) polyethylene glycol monomethyl ether 2000, 0.1 M MES monohydrate pH 6.0) for 1 min before flash-cooling to 100 K in liquid nitrogen. Diffraction datasets were collected at 100 K on BL32XU or BL41XU or BL45XU beamline of the SPring-8 (Harima, Japan), employing the automated data collection ZOO[53]. The collected datasets were processed automatically with KAMO[54]. Each dataset was indexed and integrated using XDS[55], followed by a hierarchical clustering analysis using the correlation coefficients of the normalized structure amplitudes between datasets. Finally, a group of outlier-rejected datasets was scaled and merged using XSCALE[56] or Aimless[57].

## Structure solution and refinement

General data handling was performed with the Phenix package[58]. The initial model was solved by molecular replacement using Phaser[59] with the model structure of Anc4 and Anc5 predicted by AlphaFold2[60]. The model building was performed with *Coot*[61]. Structures were refined by *Coot*[61] and Phaser[59]. The details of crystallization, structure determination, the data collection, data processing, and refinement statistics are given in Supplementary Tables 2 and 3. Structural figures are described and rendered by the PyMOL Molecular Graphics System, Version 1.2r3pre, Schrödinger, LLC.

## Molecular dynamics simulations

The MD simulations were performed using GROMACS version 2020.1[62] and the charmm36-mar2019 force field[63] was used for its proven accuracy in reproducing the structural and dynamics properties of similar protein systems as NMR and crystallography data. The model structures were built using Anc4 and Anc5 structures obtained from this work and used as a starting point for MD simulations. For each model, the system was solvated in a dodecahedral box using three points charge water model (TIP3P)[64] and neutralized with $Na^+$ and $Cl^-$ ions. Energy minimization was conducted using the steepest descent algorithm up to 1000 steps with a force tolerance of 10 kcal mol$^{-1}$ nm$^{-1}$. The systems were equilibrated in the NVT ensemble for 1000 ps with position restraints on the proteins, ensuring temperature stabilization at 310 K using a modified Berendsen thermostat[65]. Subsequent NPT equilibration was performed to stabilize pressure at 1 bar using the Parrinello–Rahman barostat[66]. LINCS algorithm was applied to constrain all protein bonds. Short-range electrostatics and van der Waals interactions were treated with a cut-off of 1.2 nm, while long-range electrostatics were calculated using the Particle Mesh Ewald method. Following equilibration, production MD simulations were conducted for 200 ns, repeated four times to ensure statistical reliability. The simulations utilized a 2 fs timestep with a Verlet cut-off scheme for neighbor searching, updated every 20 steps. The system set up is detailed in Supplementary Table 7. In our study, the focus is to compare the flexibility of the protein conformations and to confirm that these properties are related to protein's properties such as antifungal activity and binding activity. We have determined that our 200 ns conventional MD simulations are sufficient to analyze the fluctuation by using Gromacs standard analysis package.

## Rigidity-based allosteric communication and FIRST

Allostery analysis was carried out by applying RTA analysis[25]. The RTA method utilizes graph and rigidity theory[26,27] techniques to identify allosteric networks within structures of proteins and protein complexes[28–31]. Starting with protein structures of Anc4 + Loop II and Anc5ΔLoopI, we applied the RTA algorithm by sequentially perturbing rigidity of individual residues and monitoring changes in conformational degrees of freedom in Loop II. Rigid cluster decomposition and dilution plots on Anc4 + Loop II and Anc5ΔLoopI were performed with software FIRST[32]. FIRST creates a geometric molecular framework, whose underlying network (graph) contains atoms (vertices) and edges (i.e., constraints representing covalent bonds, hydrogen bonds, electrostatic interactions, and hydrophobic contacts). Every potential hydrogen bond is assigned an energy strength in kcal/mol, and a hydrogen bond cutoff energy value was selected so that all bonds weaker than this cutoff are removed from the network. FIRST then applies the pebble game algorithm[27,67] which rapidly decomposes a protein structure into flexible and rigid regions while incrementally removing weak hydrogen bonds.

## Microscopic observations

The catalytically inactive mutants of Anc4, Anc4 + Loop I, Anc5, AncΔ5 were tagged with Alexa Fluor 488 Microscale Protein Labeling Kit (Thermo Fisher), followed by the manufacturer's protocol.

The Alexa Fluor 488 reactive dye has a tetrafluorophenyl ester moiety that reacts with primary amines of proteins to form conjugates.

10 mL of sterilized PDB (Potato dextrose broth) medium containing 10$^6$ spores/mL of T. *longibrachiatum* was incubated at 25 °C for 24 h with shaking at 300 rpm to obtain the mycelia of the fungi. The mycelia were collected by centrifugation (3000 *g*, 25 °C, 20 min) and was resuspended by PDB medium. Then, 50 μL of the mycelial suspension was mixed with 50 μL of 2 μM each protein sample solution and incubated at 25 °C for 1 h. The incubated solution mixture was washed three times with 20 mM sodium phosphate buffer (pH 7.4). The samples were observed with confocal laser scanning microscopy.

## Reporting summary

Further information on research design is available in the Nature Portfolio Reporting Summary linked to this article.

## Data availability

Atomic coordinates and the experimental structure factors have been deposited in Protein Data Bank (https://www.rscb.org) with ID 3WH1, 4J0L, 8HNE, 8HNF, 8X2V, and 8X2W for loopless type GH19 chitinase from *Gemmabryum coronatum*, loopful type GH19 chitinase from *Secale cereale*, Anc4, Anc5, Anc4+LoopII+p12K/n13H, and Anc4+LoopII +p12K/n13H/s58T/n193G/y194F/d197R, respectively. Parameters, input files, output files from our MD simulations are available for download from Zenodo (https://zenodo.org) at https://zenodo.org/records/ 10730202. Source data are provided with this paper.

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

## Acknowledgements

Financial support by Okinawa Institute of Science and Technology (OIST) is gratefully acknowledged. We thank Nobuhiko Tokuriki for the discussion on the project. We thank Toki Taira, and Ben Clifton and Erika Fukuhara for their insightful comments on this manuscript. We are grateful for the help provided by the scientific computing, the Data Analysis, and the Instrumental sections at OIST. We also thank Paolo Barzaghi from the Imaging Section of OIST for the help with microscope experiments and Stefano Pascarelli for assistance with molecular dynamics simulations. The synchrotron radiation experiments were performed at BL45XU and BL41XU of SPring-8 with the approval of the Japan Synchrotron Radiation Research Institute (JASRI) (Proposal No. 2022A2769).

## Author contributions

D.K. and P.L. designed the project. D.K. performed phylogenetic analysis, all the biochemical experiments, biophysical characterization of all the proteins and mutants, fluorescent imaging microscope experiments, and molecular dynamics simulations. A.S. performed rigidity-transmission allostery computation and the related data analysis. D.K. and P.L. wrote the manuscript with the input from A.S. This project was supervised by P.L.

## Competing interests

The authors declare no competing interests.
