## [Peer Review File · Nature Communications]

Reviewers' Comments:

Reviewer #1:

Remarks to the Author:

The manuscript entitled "Beyond the active site: The addition of a remote loop reveals a new complex biological function for chitinase enzymes" (Kozome et al.) is an excellent work that demonstrates the crucial role of a remote loop for antifungal action of GH19 chitinases. Based on the phylogenetic tree constructed from 179 GH19 sequences, five ancestral proteins (Anc1-Anc5) were selected as nodes, where loop insertions/deletions take place. Functional and structural characterization of Anc1-Anc5 and their mutants revealed that LoopII is crucial for expressing antifungal activity in GH19 chitinases. MD simulation based on the crystal structures and RTA analysis also revealed that the LoopII stabilization caused by long-range intramolecular interactions is essential to perform antifungal action. The experimental and the calculation data were sound, and the data were correctly discussed. Since the manuscript contains sufficient novelty, it is worth to be published. This reviewer suggests two minor changes listed below.

1. Fig.3C, four close-up views are included in this figure; however, the descriptions in the text (lines 139-149) only refer Fig. 3C in parentheses. It should be clearly indicated which close-up view it corresponds to. Line 141, Asp residue \diamond Asp73. Line142, Trp residue \diamond Trp78. Line145, Tyr residue \diamond Tyr76.
2. Supplementary Figs. S6A and S6B. Right panels. An explanation of the vertical and horizontal axes (hydrogen bond energy and residue number, respectively) should be inserted into these figures. Moreover, the font size of the scale numbers for individual axes should be increased. This reviewer cannot find dashed square referred in the last sentence of the figure legend.

Reviewer #4:

Remarks to the Author:

I think the authors have done a great job in responding to referee's questions. In particular, regarding the answers to reviewer 2 which I was asked to independently evaluate, the authors have carried out a significant effort in terms of adding new content, data, and revising the text. The paper is nice and interesting to read and the response to reviewer fully documented. If a little addition could be done, some recent references on activity control in enzymes by distal modifications could be added, such as doi.org/10.1016/j.str.2023.05.017; doi.org/10.1016/j.sbi.2023.102702

Reviewer #5:

Remarks to the Author:

This is an interesting manuscript by Kozome and coworkers, examining the role of a remote loop (and loop flexibility) in the evolution of chitinase enzymes. Overall, I think it is a solid work, and it has already been reviewed in detail before. My focus has as requested been on the computational aspects of the manuscript. The simulations performed are non-controversial, and the data presented supports the authors' hypothesis. My only concern, which is a major one, is the minimal amount of detail provided about simulation setup, which is not ideal for reproducibility. I am happy to sign this review with my name - Lynn Kamerlin - and suggest the authors for example examine some recent MD based publications on our group (or those of other groups) to see the level of detail we typically provide for reproducibility. I would be happy to address any queries that arise on this point. Note that I am just referring to papers from my lab as an example, obviously this is not the template the authors need to use, I am happy as long as sufficient detail is provided for

reproducibility irrespective of format.

REVIEWER COMMENTS

Reviewer #1 (Remarks to the Author):

The manuscript entitled “Beyond the active site: The addition of a remote loop reveals a new complex biological function for chitinase enzymes” (Kozome et al.) is an excellent work that demonstrates the crucial role of a remote loop for antifungal action of GH19 chitinases. Based on the phylogenetic tree constructed from 179 GH19 sequences, five ancestral proteins (Anc1-Anc5) were selected as nodes, where loop insertions/deletions take place. Functional and structural characterization of Anc1-Anc5 and their mutants revealed that LoopII is crucial for expressing antifungal activity in GH19 chitinases. MD simulation based on the crystal structures and RTA analysis also revealed that the LoopII stabilization caused by long-range intramolecular interactions is essential to perform antifungal action. The experimental and the calculation data were sound, and the data were correctly discussed. Since the manuscript contains sufficient novelty, it is worth to be published. This reviewer suggests two minor changes listed below.

Thanks to recognize the novelty of our work and our comprehensive methods to tackle the crucial role of remote loop in GH19 chitinase enzymes.

1. Fig.3C, four close-up views are included in this figure; however, the descriptions in the text (lines 139-149) only refer Fig. 3C in parentheses. It should be clearly indicated which close-up view it corresponds to. Line 141, Asp residue \diamond Asp73. Line142, Trp residue \diamond Trp78. Line145, Tyr residue \diamond Tyr76.

Response: We have made these changes.

2. Supplementary Figs. S6A and S6B. Right panels. An explanation of the vertical and horizontal axes (hydrogen bond energy and residue number, respectively) should be inserted into these figures. Moreover, the font size of the scale numbers for individual axes should be increased. This reviewer cannot find dashed square referred in the last sentence of the figure legend.

Response: We have made these changes.

New Figures S6:

A

Anc4 + Loop II

B

Anc5 Δ Loop I

D
Figure S6. Computational analysis of long-range communication and rigidity.

A and B, Rigid cluster decomposition using program FIRST of Anc4 + Loop II and Anc5 Δ Loop I at hydrogen bond energy strength cutoff -2.3 kcal/mol are mapped on 3D structures. Distinct rigid clusters are designated by different colored regions, while flexible regions are shown in gray. Red represents the largest dominant rigid cluster. Anc5 Δ Loop I has an additional rigid cluster in loop II highlighted with a dashed circle.

C and D, Rigidity profile of Anc4 + Loop II and Anc5 Δ LoopI indicated with a hydrogen bond dilution plot with program FIRST. The horizontal axis depicts the residue numbers. The vertical axis represents the current hydrogen bond energy cutoff in kcal/mol. Flexible regions are indicated as black horizontal thin lines and distinct rigid clusters are indicated by various colored blocks with red being the largest rigid cluster. As weak hydrogen bonds are removed, clusters become fragmented, and structure becomes increasingly flexible. Dashed square indicates the rigid cluster within loop II that is observed only in Anc5 Δ Loop I.

Reviewer #4 (Remarks to the Author):

I think the authors have done a great job in responding to referee's questions. In particular, regarding the answers to reviewer 2 which I was asked to independently evaluate, the authors have carried out a significant effort in terms of adding new content, data, and revising the text. The paper is nice and interesting to read and the response to reviewer fully documented. If a little addition could be done, some recent references on activity control in enzymes by distal modifications could be added, such as doi.org/10.1016/j.str.2023.05.017; doi.org/10.1016/j.sbi.2023.102702

Response: We thanks the reviewer to have evaluated our answer to reviewer 2, and we are happy that he/she is satisfied by our new data, analysis and revision of the text.

Additionally, we appreciate the reviewer's suggestion to include recent advances in distal modification for enzyme activity control. In response, we have incorporated the second reference suggested (now ref 17), which provides insights into computing allostery, a topic relevant to our study. However, we respectfully find the first reference suggested by the reviewer (doi.org/10.1016/j.str.2023.05.017) to be beyond the scope of our study. This reference primarily focuses on post-translational modifications, a topic that is not directly related to our research.

Reviewer #5 (Remarks to the Author):

This is an interesting manuscript by Kozome and coworkers, examining the role of a remote loop (and loop flexibility) in the evolution of chitinase enzymes. Overall, I think it is a solid work, and it has already been reviewed in detail before. My focus has as requested been on the computational aspects of the manuscript. The simulations performed are non-controversial, and the data presented supports the authors' hypothesis. My only concern, which is a major one, is the minimal amount of detail provided about simulation setup, which is not ideal for reproducibility. I am happy to sign this review with my name - Lynn Kamerlin - and suggest the authors for example examine some recent MD based publications on our group (or those of other groups) to see the level of detail we typically provide for reproducibility. I would be happy to address any queries that arise on this point. Note that I am just referring to papers from my lab as an example, obviously this is not the template the authors need to use, I am happy as long as sufficient detail is provided for reproducibility irrespective of format.

Response:

We thank Prof. Kamerlin for finding our work interesting and solid.

Regarding the computational part, we have added a more detailed description about simulation setup in material methods section and Supplementary Table S7. The parameters, input files, output files from our MD simulations are available Zenodo (<https://zenodo.org>) at <https://zenodo.org/records/10730202>.

New section reads:

Molecular Dynamics Simulations

The MD simulations were performed using GROMACS version 2020.1⁶² and the charmm36-mar2019 force field⁶³ was used for its proven accuracy in reproducing the structural and dynamics properties of similar protein systems as NMR and crystallography data. The model structures were built using Anc4 and Anc5 structures obtained from this work and used as a starting point for MD simulations. For each model, the system was solvated in a dodecahedral box using three points charge water model (TIP3P)⁶⁴ and neutralized with Na⁺ and Cl⁻ ions. Energy minimization was conducted using the steepest descent algorithm up to 1000 steps with a force tolerance of 10 kcal mol⁻¹nm⁻¹. The systems were equilibrated in the NVT ensemble for 1000 ps with position restraints on the proteins, ensuring temperature stabilization at 310 K using a modified Berendsen thermostat⁶⁵. Subsequent NPT equilibration was performed to stabilize pressure at 1 bar using the Parrinello-Rahman barostat⁶⁶. LINCS algorithm was applied to constrain all protein bonds. Short-range electrostatics and van der Waals interactions were treated with a cut-off of 1.2 nm, while long-range electrostatics were calculated using the Particle Mesh Ewald method. Following equilibration, production MD simulations were conducted for 200 ns, repeated four times to ensure statistical reliability. The simulations utilized a 2 fs timestep with a Verlet cut-off scheme for neighbor searching, updated every 20 steps. The system set up is detailed in Supplementary Table S7. In our study, the focus is to compare the flexibility of the protein conformations and to confirm that these properties are related to protein's properties such as antifungal activity and binding

activity. We have determined that our 200 ns conventional MD simulations are sufficient to analyze the fluctuation by using Gromacs standard analysis package.

Supplementary Table 7. Systems used for MD simulations.

Proteins	Total atoms	Water	Ions	Dodecahedral Box size (nm)									
Anc4	159156	51960	5 Na ⁺	13.08	13.08	9.25	0.00	0.00	0.00	0.00	0.00	6.54	6.54
Anc4+Loop II	156054	50872	6 Na ⁺	13.01	13.01	9.20	0.00	0.00	0.00	0.00	0.00	6.50	6.50
Anc5	166729	54380	Na ⁺	13.30	13.30	9.41	0.00	0.00	0.00	0.00	0.00	6.65	6.65
Anc5ΔLoop I	155306	50606	2 Na ⁺	13.00	13.00	9.18	0.00	0.00	0.00	0.00	0.00	6.50	6.50
Anc4+Loop II+A	156199	50917	5 Na ⁺	13.02	13.02	9.20	0.00	0.00	0.00	0.00	0.00	6.51	6.51
Anc4+Loop II+B	156120	50893	6 Na ⁺	13.01	13.01	9.20	0.00	0.00	0.00	0.00	0.00	6.50	6.50
Anc4+Loop II+C	158165	51575	4 Na ⁺	13.06	13.06	9.24	0.00	0.00	0.00	0.00	0.00	6.53	6.53
Anc4+Loop II+A+B	156115	50888	5 Na ⁺	13.00	13.00	9.18	0.00	0.00	0.00	0.00	0.00	6.50	6.50
Anc4+Loop II+A+C	156204	50918	3 Na ⁺	13.01	13.01	9.20	0.00	0.00	0.00	0.00	0.00	6.50	6.50
Anc4+Loop II+B+C	158147	51568	4 Na ⁺	13.06	13.06	9.24	0.00	0.00	0.00	0.00	0.00	6.53	6.53
Anc4+Loop II+A+B+C	155205	50584	3 Na ⁺	12.98	12.98	9.17	0.00	0.00	0.00	0.00	0.00	6.49	6.49
Anc5 P204n/D205p	165120	53849	Na ⁺	13.24	13.24	9.36	0.00	0.00	0.00	0.00	0.00	6.62	6.62
Anc5ΔLoop I P204n/D205p	155086	50530	0	12.98	12.98	9.18	0.00	0.00	0.00	0.00	0.00	6.49	6.49

Reviewers' Comments:

Reviewer #5:

Remarks to the Author:

The computational methodology is much better in terms of detail, and I commend the authors for creating a Zenodo data package to supplement this, which is really important for reproducibility. I am satisfied with the revisions to the manuscript, and think this is a great paper!